# DOIDS: An Intrusion Detection Scheme Based on DBSCAN for Opportunistic Routing in Underwater Wireless Sensor Networks

**DOI:** 10.3390/s23042096

**Published:** 2023-02-13

**Authors:** Rui Zhang, Jing Zhang, Qiqi Wang, Hehe Zhang

**Affiliations:** 1College of Software and Communications, Tianjin Sino-German University of Applied Sciences, Tianjin 300350, China; 2College of Electronic Information and Automation, Tianjin University of Science & Technology, Tianjin 300222, China; 3School of Electrical and Information Engineering, Tianjin University, Tianjin 300072, China

**Keywords:** underwater wireless sensor networks, opportunistic routing security, intrusion detection scheme

## Abstract

In Underwater Wireless Sensor Networks (UWSNs), data should be transmitted to data centers reliably and efficiently. However, due to the harsh channel conditions, reliable data transmission is a challenge for large-scale UWSNs. Thus, opportunistic routing (OR) protocols with high reliability, strong robustness, low end-to-end delay, and high energy efficiency are widely applied. However, OR in UWSNs is vulnerable to routing attacks. For example, sinkhole attack nodes can attract traffic from surrounding nodes by forging information such as the distance to the sink node. In order to reduce the negative impact of malicious nodes on data transmission, we propose an intrusion detection scheme (IDS) based on the Density-Based Spatial Clustering of Applications with Noise (DBSCAN) clustering algorithm for OR (DOIDS) in this paper. DOIDS is based on small-sample IDS and is suitable for UWSNs with sparse node deployment. In DOIDS, the local monitoring mechanism is adopted. Every node in the network running DOIDS can select the trusted next hop. Firstly, according to the behavior characteristics of common routing attack nodes and unreliable underwater acoustic channel characteristics, DOIDS selected the energy consumption, forwarding, and link quality information of candidate nodes as the detection feature values. Then, the collected feature information is used to detect potential abnormal nodes through the DBSCAN clustering algorithm. Finally, a decision function is defined according to the time decay function to reduce the false detection rate of DOIDS. It makes a final judgment on whether the potential abnormal node is malicious. The simulation results show that the algorithm can effectively improve the detection accuracy rate (3% to 15% for different scenarios) and reduce the false positive rate, respectively.

## 1. Introduction

Underwater Wireless Sensor Networks (UWSNs) are mainly composed of ground base stations, surface sink nodes, and underwater sensor nodes, which can complete military and civilian applications such as ocean monitoring, resource exploration, and target tracking. They are important technologies for countries to develop and utilize marine resources. Faced with a massive increase in the types and volumes of data, UWSNs need to transfer data to the data center efficiently, reliably, and securely. 

However, the propagation speed of an underwater acoustic signal is about 1.5 × 10^3^ m/s, which is five orders of magnitude smaller than the existing radio signal with a long propagation delay (3 × 10^8^ m/s). Compared with radio communication, underwater acoustic communication has serious temporal and spatial uncertainty. Node positions change dramatically due to factors such as wind, waves, and ocean currents, which increase the probability of packet loss caused by packet conflict. Moreover, underwater acoustic channels have a serious multipath and Doppler effect, high transmission loss, and high environmental noise, which also causes high error rates and high packet loss, to some extent. These features pose great challenges for reliable data transmission [1,2].

In this regard, opportunistic routing (OR) protocols have been proposed to solve the problem of the poor and variable quality of underwater acoustic channels and to improve data transmission in UWSNs [3,4]. OR makes full use of the broadcast features of UWSNs by forwarding packets to a group of nodes (candidate node set) at a time, from which the node with the highest priority is selected as the relay node according to a specific algorithm. Therefore, OR can effectively improve the reliability and energy efficiency of UWSNs in the sparse sensor node deployment environment, and alleviate the routing void problem. However, the open underwater acoustic communication environment and unsupervised characteristics make sensor nodes in UWSNs vulnerable to malicious attacks, such as a sinkhole attack. It tampers or disrupts routing by attracting traffic so that packets cannot reach the sink node through the shortest path, thus affecting the efficiency and security of data transmission. Therefore, research on secure data transmission is very important for UWSNs.

Many security mechanisms proposed for UWSNs can effectively defend against malicious attacks. Key management and identity authentication can effectively prevent the disclosure of sensitive information and prevent malicious nodes from invading the network [5,6]. As the second protective wall, the intrusion detection scheme (IDS) can effectively address the failure of the prevention mechanism and help networks to identify internal attackers. Intrusion detection research on Terrestrial Wireless Sensor Networks (TWSNs) has gained increasing attention [7]. However, these methods cannot be directly applied to UWSNs due to the complexity of underwater acoustic channels (serious multipath effect, large environmental noise, large time delay, etc.) and the hardware equipment limitations of underwater nodes. Therefore, the design of IDS should be based on the characteristics and limitations of UWSNs, and consider the communication, memory, and energy consumption of the detection system, as well as the detection accuracy. Moreover, suitable detection schemes for different network functions should be researched.

In the past decade, the research of IDS based on trust management [7,8,9,10,11,12,13] in UWSNs has achieved some achievements; however, there are still some defects. (1) The weight of each piece of trust evidence in the trust management mechanism based on the weighting algorithm is usually determined empirically, so it cannot be assumed to be optimal. (2) IDS based on a supervised learning classification algorithm [9] can effectively distinguish between legitimate nodes and malicious nodes, while such methods require sufficient training samples. In view of the scarcity of relevant data sets of UWSNs, they can only be stuck in the simulation stage. (3) Traditional schemes usually consider that malicious nodes can only perform one attack mode. This paper considers a hybrid attack mode, which is different from the traditional attack scheme. A hybrid attack is a malicious node embedded with multiple attack modes, and can switch between the two over a period of time.

Therefore, in order to remedy the defects of the above detection mechanism and enhance the data transmission security of OR in the distributed structure of UWSNs, this paper proposes an IDS based on DBSCAN, an unsupervised clustering algorithm in machine learning [14]. Firstly, the energy consumption and forwarding rate of nodes are selected as indicators according to the behavior characteristics of common routing attacks. However, in the actual underwater scene, environmental noise can affect the judgment of node types and produce swamping and masking effects. For example, changes in the underwater environment may create weak acoustic communication links, resulting in low-quality interaction between sensor nodes, and unstable energy consumption and forwarding rates. Therefore, link quality information is taken as the third indicator. Second, the nodes in the candidate forwarding set work in similar environments and tasks, so their performances are somewhat similar. According to this feature, the majority of legitimate node samples are clustered by the DBSCAN clustering algorithm, and a few abnormal samples (noise points) caused by the malicious behaviors of nodes or weak link connections are excluded from the cluster, so as to identify abnormal samples. Then potential malicious nodes and channel abnormal nodes in abnormal samples are distinguished according to the underwater acoustic link information. Finally, in order to reduce the impact of the DBSCAN false positive rate on DOIDS detection accuracy, this paper defines a decision function according to the time decay function to determine whether the node with potential malicious behavior is a malicious node. The final analysis shows that our work is helpful in reducing the negative impact of routing attacks on OR-based data transmission.

The main contributions and novelty of this paper include:Based on the underwater OR protocols, we design a DBSCAN-based IDS to enhance the security of data transmission in UWSNs. The DBSCAN clustering algorithm is an unsupervised learning algorithm that can be used for anomaly detection without sufficient and complete training samples. It is suitable for complex and changeable underwater environments.We defined a novel decision function according to the time decay function to make a final decision on the node type (legitimate or malicious), so as to reduce the influence of the DBSCAN false positive rate on DOIDS detection accuracy.

The rest of this paper is organized as follows. Section 2 investigates and analyzes previous work on the IDS based on the trust model as well as others and the OR protocol. In Section 3 and Section 4, the DOIDS and its related models are introduced in detail. The simulation and evaluations are explained in Section 5. Finally, Section 6 concludes this article.

## 2. Related Work

This section mainly reviews the widely studied trust model-based IDS and other types of IDS in UWSNs. Some related works of OR protocols in UWSNs are also introduced. Finally, the defects of the existing IDS protocol to protect OR security are introduced.

The authors in [8] proposed an anti-attack trust model (ARTMM) based on multidimensional trust metrics to achieve accurate and efficient trust evaluation in UWSNs. ARTMM includes three trust measures, namely, link trust, data trust, and node trust. Direct trust and indirect trust are considered to improve the detection rate and reduce the false detection rate. A trust model for UWSNs based on cloud theory (TMC) is proposed in [7]. The purpose of TMC is to solve the uncertainty and ambiguity of trust and improve the accuracy of trust assessment. Moreover, TMC analyzes the packet loss layer by layer, which alleviates the influence of unreliable underwater environments and other factors on the accuracy of relevant evidence. A collaborative trust model based on SVM (STMS) is proposed in [9]. The trust prediction model was trained by a support vector machine (SVM) to evaluate the trust value accurately. The network is divided into a number of interconnected clusters, where cluster heads (CHs) and cluster members (CMs) collaborate to perform functions. The mutual supervision mechanism of two cluster heads is proposed to strengthen the security of the cluster heads. In order to ensure the security of UWSNs, ref. [10] proposed a security clustering protocol based on a weighted algorithm to calculate trust value. Only the direct trust value is considered in this protocol to avoid the extra communication energy consumption caused by transmitting the indirect trust value. In addition, the hierarchical trust management scheme is adopted, including the in-cluster node level, CH level, and sink node level. The authors in [11] analyzed the characteristics of node mobility and sound channel, and thus established an environment model that can reduce the influence of underwater environment on trust assessment. Instead of the traditional periodic update mechanism, the authors construct a trust update model to resist a dynamic attack of malicious nodes based on the environment model and reinforcement learning. In [12], the Isolation Forest algorithm is used to evaluate the reliability of sensor nodes based on the trust data set that is integrated with communication trust, data trust, energy trust, and environmental trust. The isolated forest algorithm is an unsupervised learning algorithm, which is well suited for unbalanced data sets with only a small number of negative samples. Therefore, this method is suitable for scenarios where there are a small number of malicious nodes in UWSNs and some new types of exceptions can be detected. First, SVM was used to conduct training modeling for the three types of trust evidence, and then DS evidence theory was used to fuse the three classification results in [13]. The hash underwater environment can easily misclassify normal nodes as malicious nodes. Thus, trust redemption is introduced to reduce the false detection rate by improving the trust value of nodes under an unreliable underwater acoustic channel and a weak link connection.

Although many trust model-based IDS for UWSN security have been proposed, there are still some limitations. Based on the weighted algorithm in [8,10], the determination of experience weight and trust threshold is subjective and will produce inaccurate results. The authors in [9] avoid the addition of trust measure and subjective weight, reduce the error of trust calculation, and effectively improve the detection accuracy of the defective nodes. However, when considering direct trust and indirect trust, it is difficult to distinguish credible indirect trust. The indirect trust information exchange between neighbor nodes results in an extra communication overhead. In addition, it is a challenge for UWSNs to build a real and effective data set for IDS based on supervised learning algorithms that rely on training data sets.

In addition to the trust model described above, the researchers also built IDS in other ways. The authors in [15] proposed a malicious attack detection algorithm based on DS evidence theory, which performs fusion evaluation on the suspicious probability of node temperature and packet loss rate to identify malicious nodes. However, the algorithm does not address the problem of how the neighbor node evaluates the suspicious probability of the suspicious node. The authors in [16] proposed a feature-based IDS to detect and mitigate routing attacks in UWSNs. Based on the local monitoring method, IDS is embedded in each sensor node to detect the malicious packet loss or tampering behavior of its neighbor nodes by comparing the difference between the input and output traffic. Moreover, this paper proposes a response scheme to the malicious node so that the neighbor node does not accept or send any data packets from/to the malicious node through a broadcasting alarm, thus isolating it from the whole network. However, the scheme does not consider the influence of an underwater acoustic channel on detection results. A detection technique for identifying underwater wormhole attacks is proposed in [17]. The round-trip time of a wormhole attack is shorter than that of normal links, which can be used to judge the existence of wormhole links. This method has certain requirements for a network structure and does not consider the serious spatiotemporal uncertainty of underwater acoustic channels. In [18], an anomaly location detection system in UWSNs is proposed. The use of false location estimation information will lead to the wrong location. Therefore, it is necessary to identify and ignore the relevant location data that have been maliciously tampered with. This paper designs an independent anomaly detection scheme for the sensor node and anchor node. The anomaly of a data packet sent from an anchor node to a sensor node is predicted by an auto-regression model. Due to the spatial correlation of sensor nodes, some neighboring nodes have similar movement patterns, so the anomaly index of each packet can be monitored by fuzzy logic at the anchor node.

Routing protocols are essential for efficient data transmission. Most traditional UASN geolocation routing is designed using an OR paradigm [19]. Yan et al. proposed the classical depth-based routing (DBR) [20] protocol, where the source node forwards the packet to the next hop with lower depth using the greedy criterion. The DBR protocol is based on multiple sinks, which increases PDR and reduces end-to-end latency. In [21], the authors proposed geographic and opportunistic routing based on Depth Adjustment Routing (GEDAR). Each node greedily forwards packets to the node with the lowest depth to reduce collisions. In addition, the Energy-efficient Cooperative Opportunistic Routing (EECOR) protocol is proposed in [22], which uses fuzzy rules to select the best forwarder to reduce packet collisions.

Even the most reliable OR algorithms cannot operate efficiently in the presence of routing attacks in the network [23]. For example, a sinkhole attack can disrupt the route planning of the entire network, increasing the end-to-end delay and average energy consumption of data packets transmitted from the source node to the sink node, and reducing the delivery rate. Therefore, in order to improve the security of data transmission in UWSNs and reduce the impact of routing attacks on data transmission, this paper proposes a DBSCAN-based IDS for OR in UWSNs.

Currently, there are few IDS dedicated to OR security in UWSNs. The research in [9,10,13] is designed for UWSNs based on hierarchical structures and cannot be applied to OR based on distributed structures. Other IDS proposed for distributed architectures cannot be directly applied to OR. In [7,8], packet loss rate was selected as one of the trust indicators to detect the existence of malicious packet loss when building the trust model. However, in OR protocol, even if the malicious node, which is the node with the highest priority, does not forward the packet deliberately, other nodes in the candidate node set will continue to forward the packet. In this case, the sender cannot determine whether malicious packet loss exists. Moreover, in OR, all candidate nodes receive the packet, but only the node with the highest priority forwards the packet. Therefore, ref. [16] judged that the node type is not suitable for OR by comparing whether the input and output flows are equal.

## 3. Network Model and Acoustic Propagation Model

### 3.1. Network Model

A three-dimensional heterogeneous UWSN is considered. In other words, underwater sensor nodes have different initial energy and different computing, communication, and storage capabilities. The closer the sensor nodes are to the base station, the more energy and storage space is available, so as to balance the energy consumption ratio of the whole network. As shown in Figure 1, the sensor nodes are randomly deployed underwater, and the base stations (or sink nodes) are deployed on the water surface. The packets are transmitted to the surface base station via OR. Only if the distance between two sensor nodes is within the communication radius, can they communicate, and can the sender monitor the nodes in its candidate forwarding set. We adopt a local monitoring mechanism where each node except the neighbor nodes of the sink node is embedded with DOIDS to supervise and detect nodes in its candidate forwarding set. The type of nodes is decided based on the sliding time window to avoid the attacker from being the node with the highest priority.
Assume that there are no malicious attacks in the initial stage of network deployment. This assumption is reasonable because the routing attack node is an internal attack. In order to successfully attack UWSNs, it takes a period of time for the internal attack to obtain identity authentication and become a legitimate member of the network.Assume that the number of routing attack nodes in the marine application scenario is small. In this paper, the number of malicious nodes is considered to be no more than 20% of the total number of nodes.

The node set of UWSNs can be defined as:(1)$$\;N=\left\{n_{1},n_{2},\cdots n_{i},\cdots n_{m}\right\}$$
where *n* and *m* represent sensor nodes and the number of nodes, respectively.

The candidate node set monitored by node $n_{i}$ in a monitoring slot is defined as:(2)$$\;\;C_{i}={\left\{c_{1},c_{2},\cdots c_{j},\cdots c_{k}\right\}}_{i}$$
where $c_{j}$ represents the node monitored by supervisory node $n_{i}$, and k is the number of candidate nodes. As shown in Figure 1, during data transmission, node $n_{1}$, $n_{3},$ and $n_{6}$ have next-hop nodes other than the sink, so they are supervisory nodes. Taking node $n_{3}$ as an example, within its communication range, each node that can be used as its next hop under certain conditions is a candidate node of $n_{3}$. That is, the node set $\left\{n_{5},n_{6}\right\}$ that will be defined as ${\left\{c_{1},c_{2}\right\}}_{3}$ is candidate node set of supervisory node $n_{3}$. 

This paper mainly considers the following common attacks in the process of data routing, which pose a certain threat to the security of opportunistic routing:Sinkhole attack. It can attract the traffic of the surrounding nodes by forging the distance from the surface base station, which leads to network congestion. It may forward the tampered packet information by disguising its high priority, which affects data correctness.Sybil attack. It can use multiple false identities to control or influence a large number of normal nodes in the network. For example, different identities are used to forward the same packet many times, so that the packet falls into a routing loop, increasing end-to-end delay and even causing packet loss.Hybrid attack. Malicious nodes embedded in different routing attacks can switch attack modes over a period of time. Sinkhole attack and Blackhole attack alternately, for example.On–off attack. Malicious nodes periodically launch attacks to evade detection.

### 3.2. Acoustic Propagation Model

In this paper, the Thorp model [24] is adopted to describe the underwater acoustic propagation model. The path loss of the acoustic link is defined as:(3)$$A\left(d,f\right)=d^{k}\alpha {\left(f\right)}^{d}$$
where f represents the signal frequency, and d represents the propagation distance. The spreading factor is represented by k. For spherical transmission, k=2; for cylindrical transmission, k=1; and in the actual case, k=1.5.

In addition, α(f) represents the sound absorption coefficient. The calculation method of α(f)is indicated by the empirical Thorp equation as:(4)$$\;10\log\alpha \left(f\right)=0.11\frac{f^{2}}{1+f^{2}}+44\frac{f^{2}}{4100+f^{2}}+2.75\times 10^{-4}f^{2}+0.003$$

When the length of the transmission path is *d*, the average SNR (signal-to-noise ratio) is:(5)$$SNR\left(d\right)=\frac{E_{b}/A\left(d,f\right)}{N_{0}}=\frac{E_{b}}{N_{0}d^{k}\alpha {\left(f\right)}^{d}}$$
where $E_{b}$ represents the average energy consumed to transmit a bit of data, and $N_{0}$ represents the noise power spectral density under the condition of an additive white Gaussian noise (AWGN) channel.

## 4. DOIDS

Figure 2 shows the overall framework for DOIDS, which is described in detail in this section. A complete intrusion detection mechanism includes four steps: monitoring, analysis, detection, and response. In the monitoring stage, each supervisory node in UWSNs monitors and records the energy consumption, forwarding behavior, and link quality of its candidate nodes. During the analysis phase, each node locally analyzes and processes the information collected during its monitoring. Finally, the malicious nodes are detected based on the DBSCAN algorithm and decision function. In this paper, the interference to OR is reduced by identifying malicious nodes that threaten data transmission to ensure the efficiency and security of data transmission.

### 4.1. Monitoring and Analysis

A local monitoring strategy is adopted in this paper. As shown in Figure 1, each supervisory node $n_{i}\left(i=1,2,\cdots m\right)\;$in UWSNs will actively monitor the behavior of all candidate nodes in its candidate node set, including energy consumption, forwarding, and link information. Monitoring means that the DOIDS obtain information by listening to the packets they are interested in or sending requests. For instance, the remaining energy and the number of neighbor nodes required to calculate the energy consumption information can be obtained by sending requests to each node of the candidate node set at the end of the timer of each time slot. The forwarding information can be obtained by passively listening to each candidate node for the number of times it forwards the data packet it sends. Link information can be obtained in other collaborative interactions between supervisory nodes and candidate nodes, such as handshake protocols. It analyzes the above information to detect misconduct and tries to diagnose or isolate malicious neighbors. Once malicious behavior is detected, the monitoring node sends an alert message to its neighbors to remove the node from future routes.

The task of the analysis stage is for the node $n_{i}$ to process all kinds of information collected during its monitoring and generate a data set containing three features $\left\{Feature_{energy},Feature_{forward},Feature_{link}\right\}$. The following describes how to monitor and analyze the three types of information.

#### 4.1.1. Energy Consumption Information

Candidate nodes in $C_{i}$ are the neighbor nodes of node $n_{i}$, with similar positions in space and working modes. Energy consumption $E_{j}$ over a period of time is positively correlated with the number of neighbor nodes of $c_{j}$. The more neighbor nodes $c_{j}$ has, the more links are established, and therefore, the more power is consumed. However, malicious nodes such as a sinkhole node need to forward more packets than other legitimate nodes to attract traffic. Therefore, the energy consumption of a sinkhole node is different from legitimate nodes in a candidate node set. Energy consumption of each candidate node in $C_{i}$ collected by node $n_{i}$ in a monitoring time slot includes its residual energy $E_{lastRes}$ at the end of the last time slot, residual energy $E_{res}$ at the end of the current time slot, and the number of its neighbor nodes $N_{N\;br}=\left\{n_{1}^{N\;br},n_{1}^{N\;br},\cdots n_{j}^{N\;br},\cdots n_{k}^{N\;br}\right\}$. 

In the analysis stage, taking energy consumption,

$E=E_{lastRes}-E_{res}=\left\{e_{1},e_{2},\cdots e_{j},\cdots e_{k}\right\}$ as the dependent variable and the number of neighbor nodes as the independent variable, a linear model was fitted by simple linear regression: (6)$$\;E=\beta_{0}+\beta_{1}N_{N\;br}+\epsilon$$
where $\beta_{0}$ and $\beta_{1}$ are the coefficients obtained by regression. 

$\epsilon =\left\{\epsilon_{1},\epsilon_{2},\cdots \epsilon_{j},\cdots \epsilon_{k}\right\}$ are the residuals that follow a normal distribution. Due to the abnormal energy consumption, the residual of the malicious node is different from that of other legitimate nodes, to some extent. Therefore, the residual of energy consumption information obtained by simple linear regression is taken as a feature. The energy consumption characteristics of candidate nodes $c_{j}$ in a time slot are as follows: (7)$$\;Feature_{j}^{energy}=\epsilon_{j}$$

#### 4.1.2. Forwarding Information

As the OR protocol follows an alternate mechanism to select relay nodes from the candidate set to balance energy consumption, each candidate node of $n_{i}$ acts as a relay to forward packets of similar times in a time slot. However, malicious behaviors, such as sinkhole attacks, send tampered data packets preferentially. Not only does their energy consumption differ from that of legitimate nodes, but also the number of forwarding packets of node $n_{i}$ is higher. The forwarding times of other legitimate nodes will also decrease under the influence of malicious nodes, but with similar values. During monitoring, the number of packets forwarded by candidate nodes in $C_{i}$ collected by $n_{i}$ is $T=\left\{t_{1},t_{2},\cdots t_{j,\cdots }t_{k}\right\}$. The forwarding characteristics of candidate nodes $c_{j}$ in a time slot are as follows:(8)$$Feature_{j}^{forward}=t_{j}\times k$$
where $t_{j}$ represents the number of times that $c_{j}$ forwards packets of the $n_{i}$ in a monitoring time slot, and k is the number of candidate nodes.

#### 4.1.3. Link Information

Compared with TWSNs, underwater acoustic channels have a serious multipath and Doppler effect, high transmission loss, and environmental noise, resulting in unstable underwater acoustic links. In addition, the location of underwater sensor nodes is easily affected by wind, waves, ocean currents, and other factors, leading to the interruption of communication links. Traditional IDS for UWSNs often ignores the influence of the underwater environment on node performance. For example, after the $n_{i}$ broadcasts a data packet, if the channel quality between the $n_{i}$ and $c_{j}$ is poor, $c_{j}$ cannot receive the data packet and therefore cannot become a relay node. As a result, in a period of time, the forwarding times $t_{j}$ will be lower than other nodes in $C_{i}$, and the energy consumption will also be lower. 

In order to prevent the normal node $c_{j}$ from being wrongly detected as a malicious node due to an abnormal channel, the link quality between the node $n_{i}$ and each node in $C_{i}$ is calculated. The link information collected during monitoring includes packet reception rate (PRR), signal-to-noise ratio (SNR), and link quality indicators (LQI). By establishing geometric triangular models of PRR, LQI, and SNR, the link quality corresponds to the distance between the coordinate points composed of SNR and LQI and the origin. Let us suppose that p is the number of packets sent from node $n_{i}$ to $c_{j}$, and q is the number of packets successfully received by node $c_{j}$. First, the node calculates the LQI and SNR of the packet, which are represented as $LQI_{l}$ and $SNR_{l}\left(l=1,2\cdots q\right),$ respectively. Then, according to $LQI_{l}$ and $SNR_{l}$, PRR metrics are used to calculate the window average LQI and SNR, which are expressed as: (9)$$\left\{\begin{array}{c} \bar{SNR_{W}}=\frac{\sum_{l=1}^{q}snr_{l}}{p}\\ \bar{LQI_{W}}=\frac{\sum_{l=1}^{q}lqi_{l}}{p} \end{array}\right\}$$

As shown in Figure 3, the link quality eigenvalue $Feature_{j}^{link}$ of $c_{j}$ within a time slot can be obtained by calculating the following formula based on $\bar{SNR_{w}}$ and $\bar{LQI_{W}}$.
(10)$$Feature_{j}^{link}=\sqrt{{\bar{SNR_{w}}}^{2}+{\bar{LQI_{W}}}^{2}}$$

### 4.2. Detection and Response

In the detection phase,$\;n_{i}$ performs the anomaly detection task according to the integrated data set of samples $Sample=\left\{Feature_{energy},Feature_{forward},Feature_{link}\right\}$ collected by the DBSCAN algorithm in the analysis phase. In this paper, instead of using a common classification algorithm to identify malicious nodes, the clustering algorithm is used to cluster the sample points generated by legitimate nodes and, thus, distinguish the noise points generated by abnormal nodes. The classification algorithm belongs to supervised learning, which requires training on labeled data sets. For UWSNs with severe temporal and spatial uncertainties, the classifier trained by the data set acquired in shallow water is not suitable for UWSNs in deep water. That is to say, when applying to different underwater environments, it is necessary to acquire data sets of UWSNs containing malicious nodes in different environments and mark the samples accurately. There are several challenges in acquiring data sets either by simulating the complex and changeable underwater environment or from the real environment. The clustering algorithm is an unsupervised learning algorithm, which needs no training and can adapt to changeable UWSNs. Moreover, compared with the classical K-means clustering algorithm, DBSCAN can detect malicious nodes with mixed attack mode without the need to predefine the number of divided clusters. The detection processes based on DBSCAN are as follows:

#### 4.2.1. Preprocessing of the Data Set

Due to the unstable underwater acoustic communication, the collected eigenvalues may have some default values, so the data set needs to be pre-processed before clustering through the DBSCAN algorithm. All the data need to be standardized by Z-Score [25].

#### 4.2.2. Cluster through DBSCAN Algorithm

Due to the sparse deployment of nodes in UWSNs, the number of nodes in the candidate set of node $n_{i}$ is small. In order to obtain sufficient sample size, as shown in Figure 4, samples of candidate nodes collected by $n_{i}$ in a sliding time window of s time slots were put into a data set. Finally, the outlier sample point is found by the DBSCAN algorithm, and the node generating the noise point is potentially a malicious node.

#### 4.2.3. Type Decisions for Candidate Nodes

Let Tag=−1 be the symbol of outlier sample points. Firstly, the outliers caused by hostile environments should be corrected. When the tag value $Tag_{jt}$ of $c_{j}$ in $slot_{t}$ is −1, if $Feature_{jt}^{link}$ in the sample is in line with Equation (8), that is, if it is not in the one-sided confidence interval of the mean value of link feature, the link between $n_{i}$ and $c_{j}$ is regarded as abnormal, and $Tag_{jt}$ is corrected to 1.
(11)$$Feature_{jt}^{link}<\bar{Feature_{link}}-z_{\alpha }\frac{\sigma_{Feature_{link}}}{\sqrt{k\cdot s}}$$
where $\sigma_{Feature_{link}}$ is the mean square error of the eigenvalue of the link sample. When the confidence is 95%, $z_{\alpha }$ = 1.64, the standard normal distribution table is referenced. Then, the legitimacy of each candidate node is decided based on the sliding time window. As shown in Figure 5, according to the tag value obtained by clustering in the current decision window, Equations (12) and (13) are used to determine whether the node is legitimate or not.
(12)$$d_{slot_{i}}=\{\begin{array}{c} 1+\overset{slot_{i}}{\underset{t=1}{\sum }}e_{t},d_{slot_{i}}<1\\ 1,else \end{array}$$
(13)$$e_{t}=\{\begin{array}{c} -e^{-\alpha_{1}\left(slot_{i}-t+l_{1}\right)},T\;ag=-1\\ e^{-\alpha_{2}\left(slot_{i}-t+l_{2}\right)},T\;ag=1 \end{array}\;$$
where $slot_{i}$ is the *i*-th time slot in a decision window, $e_{t}$ is the decision value of time slot $t=\left(1,2\cdots ,slot_{i}\right)$ in the current decision window, and the initial value is 1. is the reward and punishment value of time slot. The reward value is positive and the punishment value is negative. If $d_{slot_{i}}<0\;$exists in a decision window, the node is judged to be malicious. Parameters α and l can be obtained from the time decay function (14):(14)$$\;N\left(t\right)=e^{-\alpha \left(t+l\right)}\;$$

Assuming that the attenuation function attenuates from $N_{init}\;$and after m time slots to $N_{finish}$, equations can be written as follows: (15)$$\{\begin{array}{c} e^{-\alpha l}=N_{init}\\ e^{-\alpha \left(l+m\right)}=N_{finish} \end{array}\;$$

The parameters α and l can be obtained by solving Equation (15). The selection of $N_{init}$ and m depends on the degree of reward and punishment for clustering points or outlier points. For example, when the decision value of candidate node $c_{1}$ is calculated in the current time slot $slot_{i}$, if$\;Tag_{1i}=-1$, that is, node $c_{1}\;$has potential malicious behavior in$\;slot_{i}$, it will be punished. $N_{init}=0.5$ can be achieved, i.e., the punishment value in $slot_{i}\;$is −0.5, indicating that the potential malicious behavior of node $c_{1}\;$in $slot_{i}\;$reduces its decision value by 0.5. The larger m is, the slower the time function decays, and the slower the penalty value caused by the potential malicious behavior of history decreases. In order to make the punishment before m time slots have no influence on the decision in the current time slot, let $N_{finish}=0$. However, let $N_{finish}=10^{-4}$ to make the formula solvable. When parameters are selected according to Table 1, the change in decision value d with T ag in each time slot of the decision window is shown in Figure 6. It can be seen from the figure that for continuous or periodic attacks, d will be less than 0 after a period of time. Moreover, even if there is an occasional misjudgment on the node sample, the influence on its decision value decreases over time.

Finally, the responses to malicious nodes include alerts or quarantining. Isolation response was adopted in this paper; that is, when a node judges a candidate node as a malicious one, it will not be included in the candidate node set. In other words, the malicious node cannot continue to become the routing relay node for that node. In addition, node $n_{i}$ should delete the malicious samples whose Tag=−1 in its memory, to prevent these noise points from influencing the detection results through DBSCAN clustering later. 

## 5. Simulation Results and Analysis

In this section, our work’s performance is evaluated from four aspects. First, the effects of different MinPts and Eps values on DOIDS were compared. Second, a set of appropriate parameters is selected to compare the average hop count, delivery rate, and energy tax of the network under a normal, malicious attack without IDS, and malicious attack with IDS. Then, we simulate and analyze the defense effects of DOIDS under different types of attacks. Finally, the detection rate and false detection rate of IDS and SARA against route attacks are compared.

### 5.1. Simulation Setting

We evaluate and compare the performance of DOIDS. The simulation parameters are listed in Table 2, and opportunistic routing in [26] is adopted. We randomly deployed 100 nodes in a [5500 × 5500 × 5500] $m^{3}$ 3-D network. As shown in Figure 7, the sink node is deployed at the purple pentacle, and the points in different colors represent different hops to the sink node. Each node (the node directly connected to the sink node can be ignored) is embedded with DOIDS to monitor the energy consumption, forwarding, and link information of each node in its candidate node set. The parameters of the hardware setting listed in Table 2 for underwater communication systems are obtained from [27].

The employed performance evaluation metrics are:The true positive rate (TPR, %): Ratio of the number of malicious nodes detected to the total number of malicious nodes in the candidate node set of all nodes. The larger the TPR, the higher the detection rate of malicious nodes.The false positive rate (FPR, %): Ratio of the number of normal nodes that have been misjudged to the total number of normal nodes in the candidate node set of all nodes. The smaller the FPR, the lower the false detection rate of malicious normal nodes.The Area Under Curve (AUC): It is a generic classification metric that is also used to evaluate node detection capability within IDS models. The closer the AUC is to one, the better the IDS is at distinguishing between normal and malicious nodes. The specifics of the metric are described in [28].

### 5.2. The Performance under Different MinPts and Eps

First, we compare the effects of MinPts on TPR, FPR, and AUC when there were 5% or 20% of the malicious nodes in the network. The value of MinPts should ensure that the sample points generated by malicious nodes will not become core points or boundary points; that is, the value of MinPts should not be less than the total number of samples generated by malicious nodes in a data set. As shown in Figure 8a, this paper sets that the node can be judged as a malicious node when three malicious samples appear consecutively or at intervals. When two malicious nodes appear simultaneously in a candidate node set, the number of malicious samples in a detection window is at least six. Therefore, when MinPts = 4, or 5, malicious samples will become core points or boundary points, resulting in a low detection rate. However, as shown in Figure 8b, the larger the MinPts, the fewer normal samples become core points, and the more samples become boundary points and outliers, thus increasing the recognition rate (TPR) of malicious nodes. However, the probability of misjudging normal nodes as malicious nodes (FPR) will also increase correspondingly; that is, AUC will decrease with the increase in MinPts.

Then, different Eps also affect TPR, FPR, and AUC. As shown in Figure 9a, the larger the Eps, the more samples that can become core points, and the higher the probability of malicious nodes becoming boundary points. Therefore, the identification rate (TPR) of malicious samples is reduced, while the probability of legitimate nodes being misjudged as malicious nodes is correspondingly reduced.

In addition, it can be seen from Figure 8 and Figure 9 that the detection method proposed in this paper has a better detection effect with a small number of malicious nodes in the network. The main reason is that IDS is based on outliers in the DBSCAN clustering algorithm. The pseudo-code of the DBSCAN algorithm is shown in Algorithm 1. Outliers are characterized as such when they are far away from most other sample points in the data set and the number of samples close to them is small. However, with the increase in malicious nodes in the network, the number of malicious nodes in the candidate nodes of each node increases, resulting in the multiple growth of the number of malicious samples. Therefore, the chances of malicious samples becoming core and boundary points increase, which ultimately leads to the greatly reduced recognition rate of IDS for malicious nodes.
**Algorithm 1**: DBSCAN algorithm pseudo-code
**Input:** Sample set $D=\left\{x_{1},x_{2},\cdots \cdots ,x_{m}\right\}$, neighborhood parameters (Eps,Minpts)
**Workflow:**Initializing the core object set: Ω=∅**;****for***j* = 1, 2……, *m*
   Determine the number of samples $N_{Eps}\left(x_{j}\right)$ in the Eps-neighborhood of sample $x_{j}$**;**   **if**
$N_{Eps}\left(x_{j}\right)\geq Minpts$
     Put sample $x_{j}$ into the core sample set: $\Omega =\Omega \cup \left\{x_{j}\right\}$**;**  
**end if**
**end for**Initialize the number of clusters: k=0**;**Initialize the set of unaccessed samples: Γ=D**;****while**Ω≠∅     Sample sets not currently being accessed:$\Gamma_{old}=\Gamma$**;**     Randomly select core objects c∈Ω, initializing the queue Q=〈c〉**;**     *Γ* = *Γ*  \ {c}**;**     **While**
Q≠∅
       Take the first sample *q* from the queue Q
       if $N_{Eps}\left(x_{j}\right)\geq Minpts$
           Make $\Delta =N_{Eps}\left(q\right)\cap \Gamma$**,**           Put all elements of Δ into the queue Q;           *Γ* = *Γ* \ Δ**;**       **end if**

    **end while**

    *k* = *k* + 1, generate clusters Ck=Γold \ Γ**;**
    Ω=Ω\Ck**end while****Output:** cluster classification C={C1,C2,⋯,Ck}**;**

### 5.3. Comparison of Network Performance

In order to achieve better detection performance when 5% and 20% of the malicious nodes exist in the network, MinPts = 9 and Eps = 1.2 were taken as parameters by analyzing and comparing the influence of parameters MinPts and Eps on DOIDS detection performance. This part compares the average hop count, packet delivery ratio, and energy tax of UWSN packets transmitted to the sink node under normal conditions, malicious attack without DOIDS, and malicious attack with DOIDS, respectively. In this section, the malicious attack is the sinkhole attack.

Average Hop Count: The average hop count is the average number of relay nodes needed to route a packet from the source node to the sink node. Ideally, packets should always be able to route to the sink node using the shortest path with the fewest relay nodes.Packet Delivery Ratio: PDR is defined by the ratio of the number of packets received by the Sink node to the number of packets sent by the source node:

(16)$$\;\;PDR=\frac{R_{packets}}{S_{packets}}\;$$
where $R_{packets}$ represents the number of packets received by the sink node, and $S_{packets}$ represents the number of packets sent by the source node.

Energy Tax: The energy tax is the average energy consumed by each node to route a packet towards the sink node, including the energy consumption of receiving and sending packets as well as the energy consumption in an idle state.

(17)$$EnergyTax=\frac{E_{comsumed}}{m\times R_{packets}}\;$$
where $E_{comsumed}$ and m represent the total energy consumed during a round of working and the number of deployed nodes, respectively.

Figure 10 compares the influence of DOIDS on the average hop count, packet delivery ratio, and energy tax of network packet routing to the sink node when there are different proportions of malicious nodes in the network. As can be seen from the figure, the existence of malicious nodes increases the average hop count and energy tax, and reduces the packet delivery ratio. By embedding IDS for each node in the network and isolating the malicious node after it is detected by the system, the impact of malicious attacks on network performance can be reduced to some extent; although, the performance cannot be completely restored to the network state without malicious nodes. The reason is that isolating a node is equivalent to no longer using it as a routing relay node. The larger the proportion of malicious nodes, the fewer nodes can be used as relay nodes in the network. As a result, the network cannot be restored to the no-attack state. However, the overall performance of the isolated response is still better than that of the malicious node, indicating that the method in this paper is effective in defending against routing attacks.

### 5.4. Performance Evaluation of DOIDS under Different Routing Attacks

In this section, we simulate and analyze the defense effects of DOIDS under different types of attacks. As shown in Table 3, we tested the impact of five different types of routing attacks on the average hop count of delivered packets, packet delivery ratio, and energy tax respectively. Refer to Section 3 for detailed descriptions of these types of routing attacks.

We compared the network performance of UWSNs with 10% of malicious nodes before and after the DOIDS response. Refer to Figure 10 in Section 5.3, when no malicious nodes exist on the network, the average packet hop count is 3.37, packet delivery ratio is 96.16%, and energy tax is 6.48 J. Obviously, several types of attacks have different degrees of negative impact on network performance. This is because the Sybil attack can use its multiple identities to repeatedly forward the same packet that falls into the routing loop. The simulation in this paper sets Time to Live as 15 hops; that is, the packet is automatically discarded when the forwarding times exceed 15. Therefore, Sybil attacks can significantly reduce the packet delivery ratio. In addition, the average packet hop count is slightly affected because the discarded packets cannot reach the sink node. The generation of routing loop will increase the forwarding behavior of nodes in the network; so for each packet, the energy tax to reach the sink node will also increase. Thus, we can conclude that Sybil attacks can seriously disrupt normal routing planning. However, under the detection and response of DOIDS, the performance of the network with Sybil nodes is improved. Because of the isolation response in this paper, network performance cannot be restored to a situation without malicious nodes.

Packets forwarded by the sinkhole node are not easy to fall into the routing loop, but they will disturb the routing because of their routing attraction. Therefore, the existence of sinkhole nodes results in a slightly lower delivery ratio and increases the average hop count and energy tax of delivered packets. A hybrid attack is the alternation of the above two kinds of attacks, with an impact on the network somewhere in between. An on–off attack is a periodic attack that starts and stops at an equal time. They will obviously have less impact on the network than a persistent attack.

At last, according to Table 3, when 10% of malicious nodes exist in UWSNs, DOIDS has similar response effects on these types of routing attacks, indicating that DOIDS is effective in defending against them. DOIDS can improve network performance after routing attacks.

### 5.5. Comparison of Schemes against Routing Attacks

The SARA [16] is selected as a comparison algorithm, which is a secure scheme against routing attacks. In the SARA algorithm, the local monitoring strategy is also adopted. However, the difference is that SARA detects malicious nodes by comparing whether the input and output traffic of the monitored nodes are equal. The DOIDS proposed in this paper combines three measures to improve detection accuracy and considers channel information to reduce the false detection rate.

As shown in Figure 11, it can be concluded that when encountered by a routing attack, e.g., a sinkhole attack, the proposed DOIDS algorithm is more accurate for identifying malicious nodes, and the false detection rate is lower. The main reason for this is that SARA does not consider the influence of packet loss caused by underwater acoustic channel conditions on the input and output flow matching. That is, the flow input and output of normal nodes are also prone to mismatch due to channel fluctuation. In order to ensure that the false detection rate is acceptable, the identification rate of the routing attack node is also reduced.

However, in terms of the decreasing trend of TPR, the scheme proposed in this paper is faster. It is not suitable for many malicious nodes in the network based on the above analysis.

In the second experiment, we evaluate the change in detection accuracy and false detection rate of four trust models over time in the mixed attack mode. The model mentioned in this paper is compared with ARTMM, TMC, and LTrust, which are all proposed trust models for UASN. The proportion of malicious nodes is set to 25% and randomly distributed in the network. In Figure 12a the detection rate of each trust model is low at the beginning, and gradually increases and stabilizes over time for all trust models. The DOIDS model, with its filtering defense mechanism, separates false recommendations from dishonest recommendations according to the quality of the underwater acoustic link, and punishes dishonest nodes accordingly, reducing the impact of malicious attacks on network performance. Therefore, its detection accuracy rises quickly and always stays at a high position. At the beginning of the simulation, due to the lack of trust evidence, LTrust does not have enough data sets to predict the trust value of the computing node, so its detection is low. As the simulation time increases, the interaction information between nodes increases, more trust evidence is available, the predicted values are more accurate, and the detection accuracy is improving. Moreover, LTrust considers the influence of the underwater environment to reduce its impact on the trust values, so its detection rate is higher than ARTMM and TMC in the later stages. Similarly, as shown in Figure 12b the DOIDS model has the lowest false detection rate and it is always within the controllable range within 10 time stages. However, the other three trust models do not consider wrong recommendations and treat all unreliable recommendations as if they were sent by dishonest nodes, which increases the false detection rate to some extent.

## 6. Conclusions

In this paper, we investigate an IDS named DOIDS for UWSNs opportunistic routing to prevent common routing attacks and to secure data transfer tasks. DOIDS adopts the local monitoring mechanism. Each node in the network monitors the energy consumption information, forwarding information, and link quality information of each candidate node in its candidate node set. Based on the DBSCAN algorithm and time decay function, the routing attack nodes in candidate nodes are identified. Finally, the simulation analysis shows that the algorithm can effectively improve the average hop count, delivery rate, and energy tax performance of packets transmitted to the sink node when there is a small proportion of malicious nodes in the network. Compared with the existing security mechanism against routing attacks in UWSNs, DOIDS has higher detection accuracy. In the future, there are still practical challenges that need to be addressed as we apply DOIDS to marine applications. First of all, our scheme has a better effect on a small number of malicious nodes. However, we should also consider how to improve the detection rate when there are a large number of malicious nodes in the network. In addition, water flow can cause dynamic motion of underwater sensor nodes in practical applications. We plan to analyze and model the mobility of underwater nodes in the future. Finally, we need to improve the scheme to implement the detection of other attack types, such as collision attacks at the MAC layer.

## Figures and Tables

**Figure 1 sensors-23-02096-f001:**
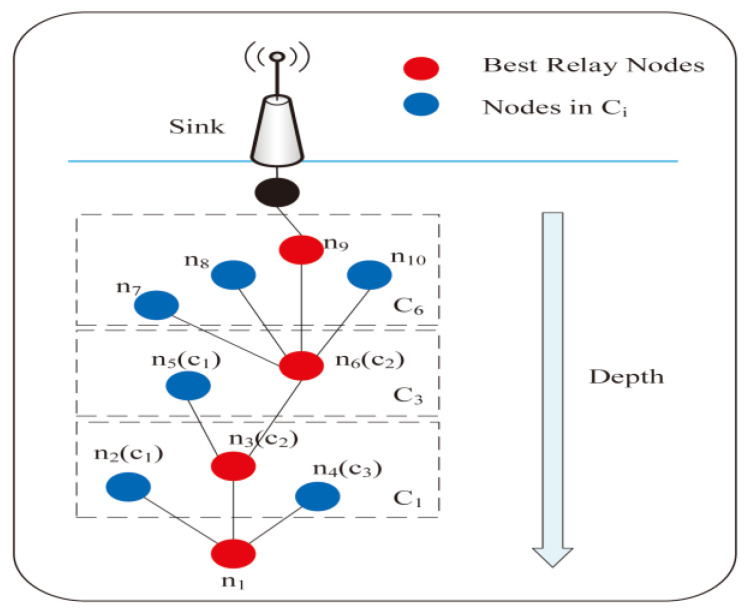
UWSN scenarios based on OR.

**Figure 2 sensors-23-02096-f002:**
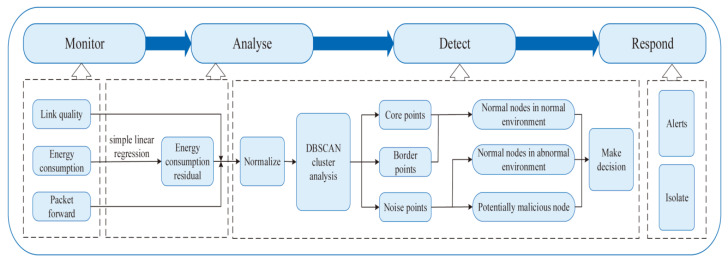
Framework for DOIDS.

**Figure 3 sensors-23-02096-f003:**
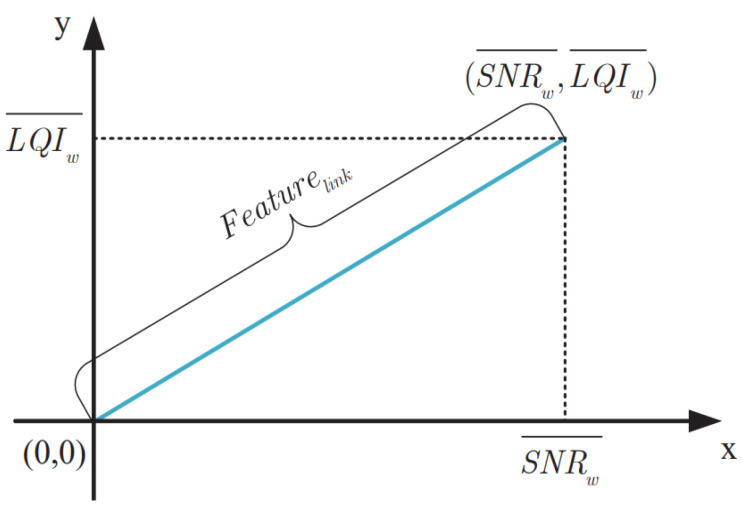
Link information Calculation.

**Figure 4 sensors-23-02096-f004:**
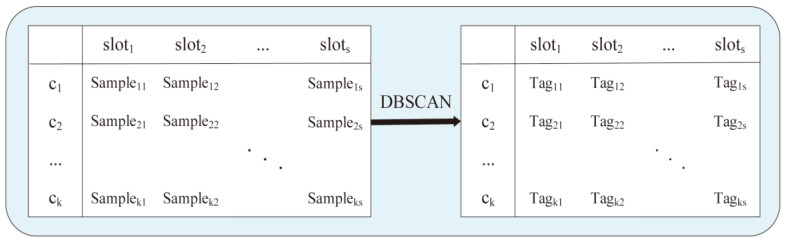
Outlier points detection based on DBSCAN.

**Figure 5 sensors-23-02096-f005:**
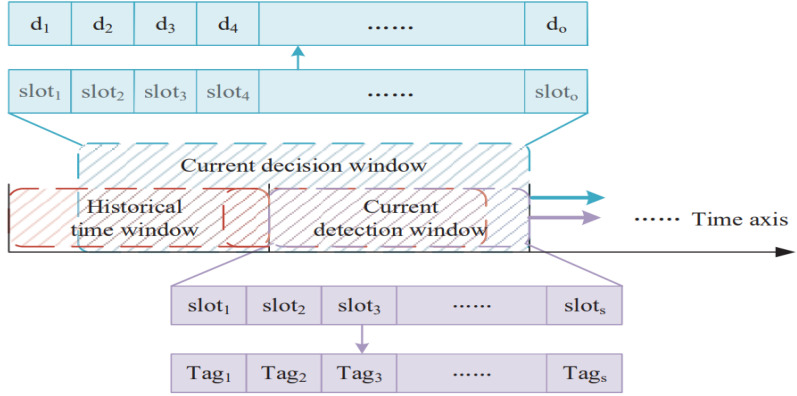
Schematic diagram of detection and decision sliding time window.

**Figure 6 sensors-23-02096-f006:**
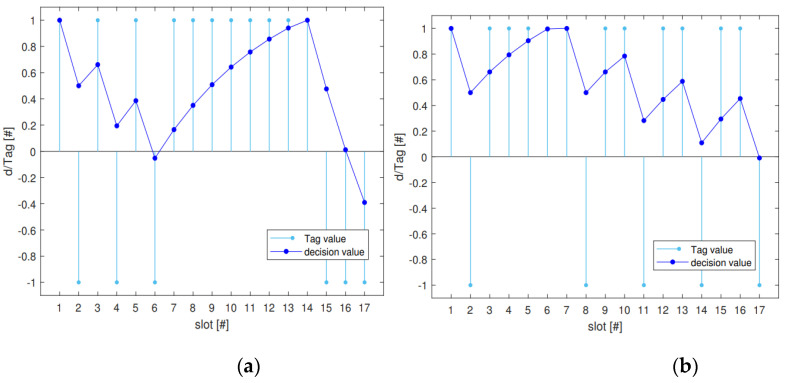
(**a**,**b**) Change curve of decision value with Tag value.

**Figure 7 sensors-23-02096-f007:**
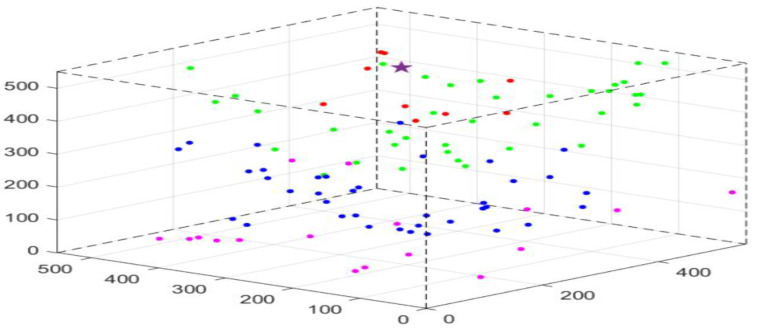
Network structure.

**Figure 8 sensors-23-02096-f008:**
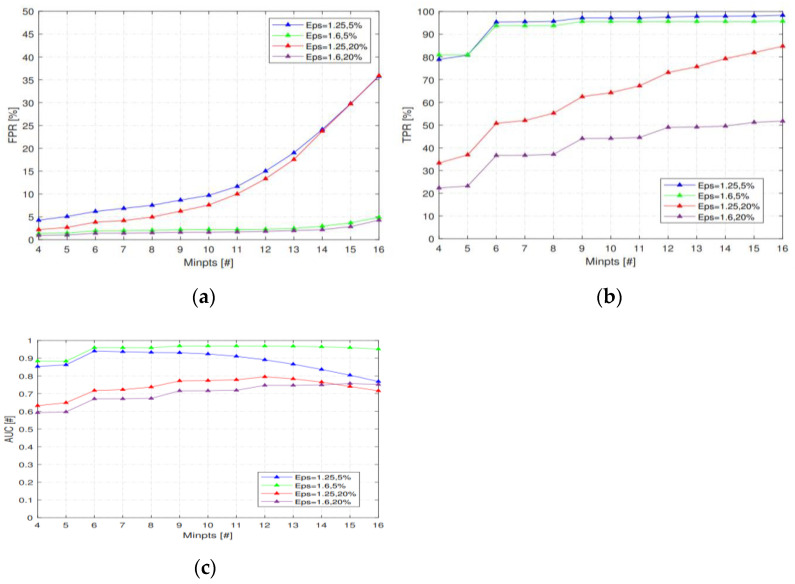
(**a**) The effects of MinPts on TPR; (**b**) The effects of MinPts on FPR; (**c**) The effects of MinPts on AUC.

**Figure 9 sensors-23-02096-f009:**
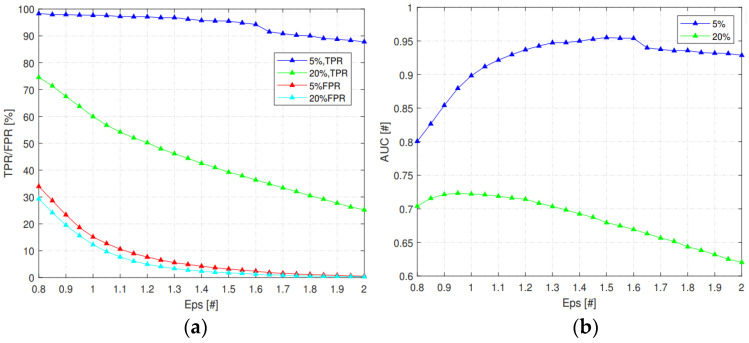
(**a**) The effects of Eps on TPR and FPR; (**b**) The effects of Eps on AUC.

**Figure 10 sensors-23-02096-f010:**
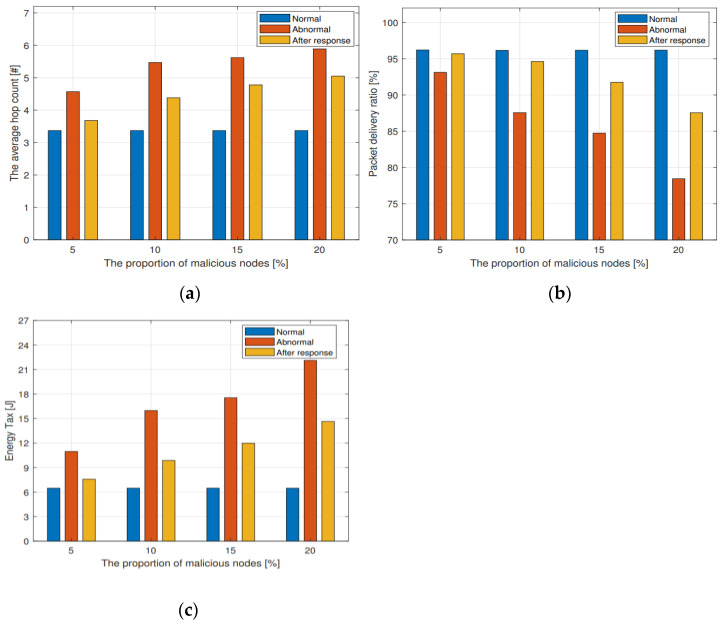
(**a**) The influence of DOIDS on the average hop count; (**b**) The influence of DOIDS on the packet delivery ratio; (**c**) The influence of DOIDS on the energy tax.

**Figure 11 sensors-23-02096-f011:**
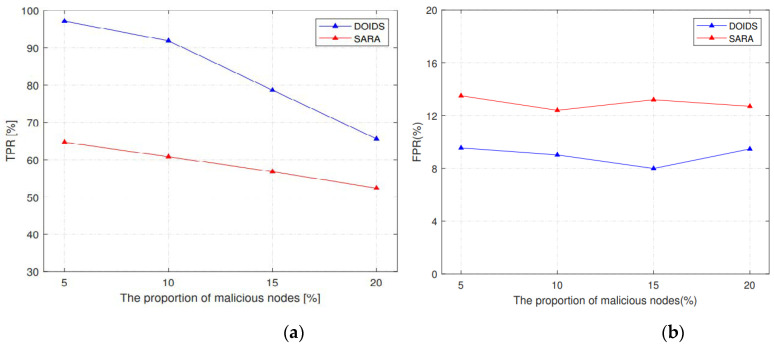
(**a**,**b**) Comparison between DOIDS and SARA.

**Figure 12 sensors-23-02096-f012:**
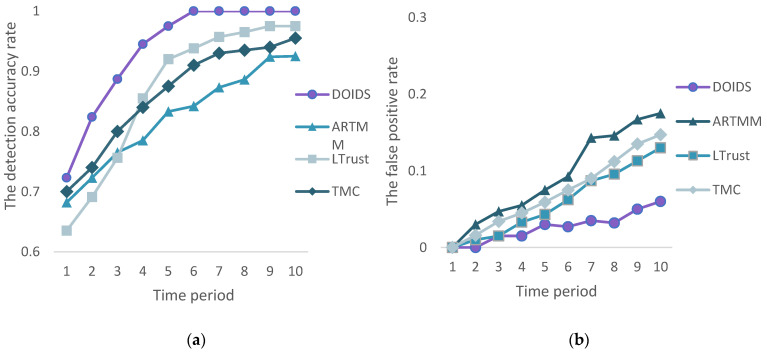
(**a**) Detection accuracy comparison; (**b**) False detection rate comparison.

**Table 1 sensors-23-02096-t001:** Parameter selection of decision function.

*T ag*	$N_{init}$	$N_{finish}$	m
−1	0.5	$10^{-4}$	30
1	0.1	$10^{-4}$	10

**Table 2 sensors-23-02096-t002:** Simulation parameters.

Name	Value
The size of UWSNs	5500 × 5500 × 5500 [${\;m}^{3}$]
The number of sensor nodes	100 [#]
Communication range	2000 [m]
Node placement Randomly deployed Acoustic channel bandwidth	20 [Kbps]
The length of detection sliding window	6 [#]
The length of decision sliding window	6 [#]
The length of packet	1024 [bit]
Data transfer rate	6 [Kbps]
Transmit mode power consumption	2 [w]
Receive mode power consumption	0.75 [w]
Time to live	15 [hops]

**Table 3 sensors-23-02096-t003:** Experimental results of 10% of malicious nodes before and after DOIDS response.

Malicious Node Type	Average Hop Count of Delivered Packets (#)		Packet Delivery Ratio (%)		Energy Tax (J)	
	Before	After	Before	After	Before	After
Sinkhole	5.47	4.38	87.65	94.61	15.98	9.87
Sybil	3.85	4.31	56.24	94.46	43.98	9.65
Hybrid	4.86	4.32	73.04	94.81	30.27	9.65
On–off (Sinkhole)	4.25	4.28	92.97	94.82	10.07	9.45
On–off (Sybil)	3.91	4.33	69.30	94.59	23.37	9.64

## Data Availability

Not applicable.

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
