# Peer review of "DOIDS: An Intrusion Detection Scheme Based on DBSCAN for Opportunistic Routing in Underwater Wireless Sensor Networks"

_sensors, 2023, doi:10.3390/s23042096_

Round 1

Reviewer 1 Report

In this paper, the opportunistic routing (OR) protocols is claimed to be the high reliability, for low end-to-end delay and high energy. The paper need clarifications on following quires

1. The abstract need to revise with results attained.

2. The comparison table of existing techniques and proposed technique need to include 

3. Validation against the existing work need to include by comparing results with results attained from past three relevant papers.

4. Selection of Parameters and its criteria need to be address in methodology.

5. What is the best-case benchmark for reduced false detection rate of DOIDS.

6. How the statement will be validated the  abnormal nodes through DBSCAN clustering algorithm? from result point of view.

Reviewer 2 Report

This paper proposed an intrusion detection scheme (IDS) based on Density-Based Spatial Clustering of Applications with Noise (DBSCAN) clustering algorithm for opportunistic routing in UWSNs to prevent common routing attacks and to secure data transfer tasks. The simulation results indicated the feasibility of the proposed algorithm. However, some concerns should be addressed before its publication.

1. In the paper, the novelty of the proposed algorithm should be addressed when compared to the existing schemes in the introduction.

2. The scales of some figures, such as fig.1 and fig. 3 should be adjusted.

3. The pseudo-code of the proposed algorithm should be given in the manuscript.

4. The packet formats of the employed packets should be given for more clear explanation.

5. Some grammar should be corrected such as “A trust model for UWSNs based on cloud theory (TMC) is proposed in [7]”, “In addition, the Energy-efficient Cooperative Opportunistic Routing (EECOR) protocol is proposed in [22], which uses fuzzy rules to select the best forwarder so as to reduce packet collisions”, “Reference [9], [10], [13] is designed for UWSNs based on hierarchical structure and cannot be applied to OR  based on distributed structure”.

Round 2

Reviewer 1 Report

The comments are addressed.

Reviewer 2 Report

The current version has been improved, and my concerns have been addressed.